# Intratumoral Distribution of Lactate and the Monocarboxylate Transporters 1 and 4 in Human Glioblastoma Multiforme and Their Relationships to Tumor Progression-Associated Markers

**DOI:** 10.3390/ijms21176254

**Published:** 2020-08-29

**Authors:** Carolin Kubelt, Sönke Peters, Hajrullah Ahmeti, Monika Huhndorf, Lukas Huber, Gesa Cohrs, Jan-Bernd Hövener, Olav Jansen, Michael Synowitz, Janka Held-Feindt

**Affiliations:** 1Department of Neurosurgery, University Medical Center Schleswig-Holstein UKSH, Campus Kiel, 24105 Kiel, Germany; Hajrullah.Ahmeti@uksh.de (H.A.); Gesa.Cohrs@uksh.de (G.C.); Michael.Synowitz@uksh.de (M.S.); Janka.Held-Feindt@uksh.de (J.H.-F.); 2Department of Neuroradiology, University Medical Center Schleswig-Holstein, UKSH Campus Kiel, 24105 Kiel, Germany; Soenke.Peters@uksh.de (S.P.); Monika.Huhndorf@uksh.de (M.H.); olav.jansen@uksh.de (O.J.); 3Section Biomedical Imaging, Molecular Imaging North Competence Center (MOIN CC), Department of Radiology and Neuroradiology, University Medical Center Schleswig-Holstein, Campus Kiel, Kiel University, 24118 Kiel, Germany; lukas.huber@rad.uni-kiel.de (L.H.); jan.hoevener@rad.uni-kiel.de (J.-B.H.)

**Keywords:** glioblastoma, MCT1, MCT4, lactate, EMT, stemness, dormancy, MRSI

## Abstract

(1) Background: Metabolic reprogramming has been postulated to be one of the hallmarks of cancer, thus representing a promising therapeutic target also in glioblastoma multiforme (GBM). Hypoxic tumor cells produce lactate, and monocarboxylate transporters (MCTs) play an important role in its distribution; (2) Methods: We examined the distribution of lactate by multi voxel magnetic resonance spectroscopic imaging and ELISA in glioblastoma multiforme (GBM) patients. In addition, we investigated the expression and cellular localization of MCT1, MCT4, and of several markers connected to tumor progression by quantitative PCR and immunofluorescence double-staining in human GBM ex vivo tissues; (3) Results: The highest lactate concentration was found at the center of the vital parts of the tumor. Three main GBM groups could be distinguished according to their regional gene expression differences of the investigated genes. MCT1 and MCT4 were found on cells undergoing epithelial to mesenchymal transition and on tumor stem-like cells. GBM cells revealing an expression of cellular dormancy markers, showed positive staining for MCT4; (4) Conclusion: Our findings indicate the existence of individual differences in the regional distribution of MCT1 and MCT4 and suggest that both transporters have distinct connections to GBM progression processes, which could contribute to the drug resistance of MCT-inhibitors.

## 1. Introduction

Glioblastoma multiforme (GBM) is the most common and most malignant brain tumor. There are still no curative treatment concepts available [1,2]. Rapid growth, despite a lack of energy substrates, and an early migration of the tumor cells, are responsible for its poor prognosis [1]. To develop these aggressive properties, the tumor cells undergo molecular changes, which can lead to a change in cell metabolism or phenotype. In the early 1920s, Otto Warburg, a German scientist and later Nobel Prize laureate, observed increased lactate levels in tumor cells. He hypothesized that tumor cells generate energy via oxygen-independent glycolysis, despite the availability of oxygen. This process, which has since been named “the Warburg effect”, and which is characteristic of many tumor entities, appears to represent an adaptation mechanism that enables tumor cells to meet the high energy requirements caused by their rapid proliferation [3]. Proton-coupled monocarboxylate transporters (MCT) have been described in connection with the distribution or membrane passage of lactate produced by oxygen-independent glycolysis. MCT1 and MCT4, in particular, are overexpressed in GBM cells [4]. Previous studies have shown a positive correlation between the expression of both transporters, with tumor cell proliferation, angiogenesis, and the degree of malignancy of gliomas [5]. Initial findings from studies focusing on the function of the transporters in animal experiments suggest that MCT4 is responsible for the exportation of lactate. On the other hand, the exported lactate could be absorbed again via MCT1 and then be metabolized for further energy production [4]. Lactate transporters also seem to play a role in the context of the epithelial to mesenchymal transition (EMT), a process for generating a highly invasive GBM phenotype [5]. Cells with a stem cell-like character, which are associated with the high invasiveness of GBM cells and their resistance to radio- and chemotherapy [6], showed an increased expression of MCT1 and MCT4 [5]. Interestingly, a close relationship between the tumor stem cell theory and the dormancy phenomenon seems to be obvious [7].

In the framework of developing new therapy strategies against GBM, inhibitors of MCT1 and MCT4 are promising targets, thus understanding their relation to other tumor progression related processes is of great interest [4,6]. Furthermore, the technique of MRSI might take an important position in this context, since it offers the possibility of gaining insights into the metabolism noninvasively. Currently, only a few studies focus on the connection between MCT proteins and other tumor progression related processes in GBM, and detailed in situ work regarding the MCT expression in human GBM is not available. Furthermore, only little is known concerning a possible connection between MCT expression and changes in multi voxel magnetic resonance spectroscopic imaging (MRSI), a method for assessing local metabolites.

Thus, the objective of this study was to compare the expression profiles of MCT1 and MCT4, as well as EMT, stem cell and dormancy markers in human GBM ex vivo tissue samples from the center and edge of the tumor, using quantitative reverse transcription PCR (qRT-PCR) and immunofluorescence staining. Additionally, the results were compared, both with the lactate concentration determined by an enzyme-linked immunosorbent assay (ELISA) in the ex vivo tissues samples, and with a multi voxel MRSI of the areas examined before surgery, in particular with regard to the lactate peak.

## 2. Results

### 2.1. Lactate Concentrations Are Higher in the Contrast-Enhancing Part of the GBM with an Accumulation at the Center of the Vital Parts of the Tumor

First, 8 GBM patients included in the study underwent multi voxel MRSI before surgery, in particular with regard to the lactate peak at 1.3 ppm.

Lactate was detected in all cases, particularly in the contrast-enhancing part of the tumor, meaning in T1-weighted images after intravenous contrast agent injection (Figure 1), but the measured lactate concentrations were different for each patient (Figure 2(a1,2); see Appendix A for absolute concentration values of all patients and Appendix A for representative spectra of all patients), with a resulting large standard deviation. As an example, the average lactate concentration for all patients for the contrast-enhancing “tumor” area was 123 mM/cm^3^, with a standard deviation of 95 mM/cm^3^. Nevertheless, although the lactate concentrations varied between patients, their relative changes among the defined regions were similar. Defining the “normal appearing white matter (NAWM)” area next to the edema as the reference point with the value 1, the mean relative lactate concentration in the contrast-enhancing part of the GBM (“tumor”) was 17.93 (SD 5.1), and 8.59 (SD 2.75) for the surrounding edema without contrast enhancement (“edema”). No lactate signal was observed in the contralateral area, as expected. In summary, lactate concentrations were significantly higher in the contrast-enhancing part of the tumor, compared with the edema or the NAWM next to the edema (*p*: 0.006). To distinguish possible differences in the lactate concentration from the center and edge of the vital parts of the tumor in our samples in more detail, we carried out a lactate ELISA after surgery, using ex vivo tissue samples of 10 different patients. Figure 2b shows the results of each paired sample. There were significantly higher lactate concentrations—two times higher on average—at the center of the tumor compared with the edge (mean c_center_: 1.5 nmol lactate/mg tissue; mean c_edge_: 0.7 nmol lactate/mg tissue; *p*: 0.004).

### 2.2. Three Main GBM Groups Were Distinguished, Based on Their RelatIve n-Fold mRNA Expression Differences in the Center and at the Edge of the Tumor

To analyze whether the data obtained in the lactate ELISA and the multi voxel MRSI were reflected by distinct MCT expression profiles, we performed qRT-PCRs to detect possible MCT1 and MCT4 mRNA expression differences between the center and the edge of the tumor in GBM ex vivo tissue samples of 15 patients. Since EMT, stemness and dormancy processes play a decisive role in the aggressive behavior of the tumor, corresponding markers for these phenomena were also examined, based on a comparison of the mRNA expression in the center and at the edge. GAPDH (glyceraldehyde-3-phosphate dehydrogenase) and TBP (TATA-box-binding protein) served as internal standard to calculate the Δ*C*_t_ values. A Δ*C*_t_ value of 3.33 corresponds to a one order of magnitude lower gene expression. The results of the qRT-PCR are shown in Figure 3. 

Considerable levels of MCT1 and MCT4, as well as EMT, stemness and dormancy markers were found in all samples of GBM tissues, irrespective of the location. GFAP, the main intermediate filament protein in mature astrocytes [8] used as a tumor marker for glial cells [9], showed the highest mRNA expression, followed by vimentin, a protein which is expressed in cells of mesenchymal origin as the main part of the cytoskeleton, and is a marker for mesenchymal cells when assessing EMT [10]. The normalized averaged ΔC_T_ values of all investigated markers comparing center and edge GBM samples were: 1.89/2.22 (MCT1), 3.89/3.88 (MCT4), −3.67/−3.33 (vimentin), −4.49/−5.11 (GFAP), 2.94/2.33 (β-catenin), 6.05/4.94 (KLF4), 5.45/4.83 (OCT4), 1.73/1.51 (Sox2), 5.09/4.29 (EPHA5), 3.83/3.79 (H2BK), −0.33/−0.61 (IGFBP5) (see Appendix A for all Δ*C*_t_ values and n-fold changes). Altogether, a significantly decreased expression of MCT1 (*p*: 0.025) and a significantly increased expression of GFAP (*p*: 0.010), β-catenin (*p*: 0.039), KLF4 (*p*: 0.035) and OCT4 (*p*: 0.026) was detectable at the edge of the tumor compared with the center. In addition, a slightly increased expression (more than 2-fold) of EPHA5 and IGFBP5 was detectable at the edge of the tumor compared with the center. Furthermore, the search for possible relations between the mRNA expression of MCT1 and MCT4 with the different EMT, stemness and dormancy markers and with the lactate concentration yielded a positive correlation between the Δ*C*_t_ value of MCT1 with the lactate concentration (corr.: 0.78, *p*: < 0.001). The gene expression of GFAP correlated negatively with the lactate concentration (corr.: −0.59, *p*: 0.001). There was no statistically significant correlation between MCT1 and MCT4 with the different EMT, stemness and dormancy markers (see Appendix A). In addition, we performed an analysis of correlation of the qRT-PCR results, with clinical-pathological data, such as age, sex, time to relapse, MGMT and 1p19q status. All included patients were IDH wild-type. Due to the small number of patients included in the study, especially patients with recurrent tumor, it was not possible to draw any conclusions out of this calculation (see Appendix A).

The qRT-PCR results of the paired samples were analyzed in more detail by computing a ratio between the gene expressions at the edge of the tumor and the center. Figure 4 is a heatmap of the ratio of expressions. A ratio of 3 or higher, meaning a 3 or higher-fold expression between the center and the edge of the GBMs, is assigned the color black; a ratio of one as a 30% grey tone and a ratio close to zero is white. With particular attention paid to the ratio of expression of MCT1, one could distinguish three main GBM groups. The first group, containing 5 patients, was characterized by a lower MCT1 mRNA expression at the tumor edge, combined with a higher expression of GFAP and individual EMT-, stemness- and dormancy-associated markers (β-catenin, KLF4, IGFBP5) at the edge of the tumor. The second group showed a slightly to moderately lower gene expression of MCT1 and MCT4 at the edge of the tumor, whereas the EMT-, stemness- and dormancy-associated markers were expressed heterogeneously. Finally, the third group revealed a heterogenous pattern of all examined markers. Interestingly, comparing the lactate concentration at the center and edge of the tumor, the first group showed the highest difference, while the second and third group exhibited a slight or moderate difference. According to these findings, there was a statistically significant correlation between the quotient of lactate concentration (concentration_edge_/concentration_center_) and the group affiliation (corr.: 0.84, *p*: 0.018).

### 2.3. Different Cell Types Are Sources of the MCT Marker Expression

Based on the qRT-PCR results, we carried out immunofluorescence double-staining, in order to determine in detail which cells might account for the expression of MCT1 and MCT4, and also to find a potential connection to the EMT, tumor stemness and dormancy phenomenon. The tumor material mostly consisted of two samples from 5 different patients; one taken from the center of the tumor and one taken from the edge. MCT1 and MCT4 were stained with EMT markers (β-catenin and vimentin), stem-like cell markers (KLF4, OCT4 and Sox2), dormancy markers (EPHA5, H2BK and IGFBP5), as well as markers for glial and endothelial cells (GFAP and von Willebrand factor (vWF)). Representative results are shown in Figure 5 and Figure 6. This analysis focused exclusively on the determination of the cell-specific expression and location of MCT1 and MCT4, since a valid quantification of both staining intensities and amounts of positively stained cells is not possible with fluorescence microscopy.

Firstly, no general differences were noticed with regard to the location from where the sample was taken (center or edge of the tumor). Interestingly, with only some exceptions, most cells were solely positive for either MCT1 or MCT4. Moreover, MCT1 clearly co-stained with β-catenin, a protein involved in the WNT signaling pathway crucial for EMT [11], whereas vimentin showed different combinations of close location, complete separation or co-staining. In the case of KLF4, co-stained cells as well as cells, which were solely positive for MCT1 or KLF4, existed within the tumor sections. For Sox2, the different markers were found separated from each other or in the same regions, but did not seem to be located within the same structures in the cells. In most of the cases, MCT1 was not found to be expressed in OCT4-, EPHA5-, H2BK- or IGFBP5-positive cells. However, in a few cases, the different markers seemed to be in a close location, but no co-staining was detected. In the case of GFAP, a filament protein identifying glial cells, cells were detected, that were positively stained for both MCT1 and GFAP, whereas others showed the expression of only one of the markers. However, MCT1 and GFAP were found in the same regions. Further, MCT1 was expressed in many vWF-positive endothelial cells.

Compared with MCT1, staining for MCT4 was detectable in lower amounts in the center and at the edge of the GBM samples, reflecting the overall lower mRNA expression level of MCT4, as detected by qRT-PCR. Contrary to MCT1, MCT4 was only found in a few of the β-catenin expressing cells. MCT4 and β-catenin could be either found co-staining or in different locations in the cell. In the case of vimentin, several cells showed an expression of MCT4. KLF4-positive cells were characterized by a co-staining with MCT4. However, cells were detected that were strictly positive for either KLF4 or MCT4. As described for MCT1, OCT4- and IGFBP5-positive cells seemed to be mainly negative for MCT4. With regard to Sox2, no co-staining was seen with MCT4, although the markers seemed to be expressed in the same regions. EPHA5-positive cells showed a co-staining with MCT4, whereas for H2BK-positive cells, staining was only observed in the same region of the section. Furthermore, GFAP-positive cells expressed MCT4, although the markers did not seem to be located within the same structures in the cells. Lastly, vWF-positive cells seemed to be negative for MCT4.

In summary, β-catenin, as well as KLF4, were expressed in MCT1-positive cells. In addition, a co-staining of MCT4 with vimentin, KLF4 and EPHA5 was seen.

## 3. Discussion

Of all the various brain tumors known to us, GBM is the most malignant and with the worst prognosis. According to the current state of knowledge, this disease cannot be cured. The main reason for this poor prognosis is the rapid tumor growth with the early invasion of individual tumor cells into the surrounding tissue [1,2]. In addition, GBM has a high resistance to the currently used therapy strategies, so that tumor cells remaining after therapy represent the source of the rapid tumor recurrences [1,2]. The tumor grows rapidly, even though there is an increasing lack of energy substrates, especially in the center of the tumor. To enable this rapid growth, the tumor appears to be able to generate energy from oxygen-independent glycolysis [3]. The resulting lactate derived from this process is now available for further energy production in tumor parts connected to vessels [11]. In line with the studies mentioned, we found a higher concentration of lactate in the contrast-enhancing part of the GBM using MRSI. Indeed, lactate accumulates in cancer cells [12], and as previously shown in animal studies, MRSI seems to be a reliable tool for monitoring therapy [13]. Instead of the often-used single voxel spectroscopy, we used multi voxel spectroscopy to obtain smaller volumes per region, and to measure different regions of the tumor and its surrounding parenchyma. As expected, the spectral quality in the voxels close to the ventricles and the skull was so poor that the corresponding data was rejected. As described above, representative voxels for each region were chosen, and a TE of 288 ms was chosen to get a better lactate signal at 1.3 ppm [14]. Nevertheless, contamination of the lactate peak with signals of lipids cannot be completely avoided, and the amount remains unclear. Previous studies described a correlation of the progression-free survival with the lactate/creatine ratio [15,16], or the distribution of metabolites with tumor subtypes [17]. Despite the comparatively low number of patients in this study, the relative changes in the lactate concentrations among the different regions were significant. In accordance with this, we were able to detect a further accumulation in the center of the vital parts of the tumor, compared with the edge as determined by a lactate ELISA. The further distribution of the accumulated lactate is generated by MCTs [18,19]. MCT1 and MCT4 were found to be overexpressed in glioma cells [18,19]. Our investigations also showed a clear expression of MCT1 and MCT4 in all GBM samples. MCT1 showed a higher expression than MCT4 in the center and at edge of the tumor, which falls in line with the ubiquitous expression of MCT1 described in the literature [20]. Interestingly, we observed a statistically significant decreased gene expression of MCT1 at the edge of the tumor compared with the center. According to the ratios of mRNA expressions of mainly MCT1 between center and edge of the tumor, three main GBM groups could be distinguished. Whereas the first group was characterized by a lower MCT1 mRNA expression at the edge of the tumor combined with a higher expression of cell-type markers connected to tumor progression at the edge of the tumor, the second group was characterized by a lower expression of MCT1 and MCT4 at the edge of the tumor, and a heterogeneous expression of the cell-type specific markers in the different regions. The last group revealed a heterogenous pattern of all examined markers. Interestingly, the relative differences in the lactate concentrations seem to vary specifically in the center and at the edge of the tumor in these groups, as well. Since GBM is known for its distinct inter- and intra-tumor heterogeneity, the observation of a wide range of different patterns is not surprising [21]. Even considering the subgroups classified by The Cancer Genome Atlas (TCGA), a single tumor could be shown to consist of a heterogeneous mixture of cells representing all of the different subgroups [21,22,23]. Since one of the hallmarks of GBM is central hypoxia, a higher degree of heterogeneity can be expected, especially at the edge of the tumor. Here, we find regions showing different levels of vascularization and oxygen conditions. Moreover, the function of MCT1 and MCT4 is still under discussion, since MCT1 has been shown to be involved in lactate uptake [4,18] and also lactate efflux [4,20]. Since MCTs allow passive transport, their functions also rely on intra- and extracellular lactate levels and pH gradient [20]. In general, previous studies have shown a complex influence of the microenvironment on the gene expression of MCT1 and MCT4. For example, hypoxia was shown to cause the up- or downregulation of MCT1 [4,24]. Furthermore, the source of the marker expression needs to be regarded, since previous studies have postulated a reverse Warburg effect. In the reverse Warburg effect, stroma cells from the microenvironment produce lactate via aerobic glycolysis, so that the lactate can be used for further energy production in tumor cells. In this context, MCT4 has been shown to serve as a lactate exporter, whereas MCT1 serves as a lactate importer [25,26,27].

In contrast to MCT1, GFAP showed a statistically significantly increased expression at the tumor edge compared with the center. Since GFAP is the principal component of intermediate filaments in astroglial cells and high-grade gliomas seem to lose GFAP expression [28], the increased expression could be caused by the lower density of tumor cells at the edge of the tumor as a zone of tumor infiltration. Our observation of a significantly increased expression of the EMT marker β-catenin at the edge of the tumor is in line with previous studies, which have shown a higher expression of β-catenin, particularly at the invasive front of GBMs [29]. In addition, an increased expression of the stem-like cell markers OCT4 and KLF4 was detectable at the edge of the tumor compared with the center. Most studies postulate the perivascular niche, particularly in the subventricular zone and the hippocampus, and the hypoxic/perinecrotic niche to be distinct regions where stem cells are enriched. However, recent studies have shown tumor stem-like cells to be located in the invasion niche, found at the tumor periphery of GBM [28,30]. In particular, GBM cells located at the resection margin were shown to proliferate rapidly, and to be more invasive than GBM cells at the center of the tumor [31,32]. With regards to the dormancy phenomenon and the slightly increased expression of the dormancy markers EPHA5 and IGFBP5 at the edge of the tumor observed in our study, the only connection published to date is an induction of MCT4 and dormancy markers by hypoxia [5,33]. The current understanding of GBM suggests that up to 50% of the tumor mass is generated by the tumor stroma. These include endothelial cells, pericytes/mesenchymal stem cells, immune cells, and glial cells [34,35]. Since cellular entities other than tumor cells can also be the source of the MCT expression, the cellular composition of the tumor at the center and edge also influences their gene expression levels. For example, immune cells have been shown to express MCT1 and MCT4 [20].

The plasma membrane location and the activity of MCT1 and MCT4 were both shown to be regulated by co-expression with the same chaperone CD147 (basigin) [36]. Thus, an expression of both markers in the same cell seems to be explainable. A co-expression of MCT1 and MCT4 has also been shown to be present in breast cancer cells [37]. In our previously published study, we observed an expression of β-catenin and vimentin in GBM cells [38]. A co-staining of both markers with MCT1 and MCT4, which we found in this study, was thus expected. Additionally, in the case of the stem-like cell marker KLF4, co-expressions were found with MCT1 and MCT4. This finding is supported by a study in which an increased lactate transport via monocarboxylate transporters (MCT1-4) was found in glioma stem-like cells [39]. Moreover, previous studies showed an upregulation of MCT4 in GBM neurospheres [5].

Our observation of an expression of MCT4, but not MCT1, in dormant cells could imply that this cell type also contributes to the reverse Warburg effect. Since dormant cells are in cell cycle arrest, they themselves have no high energy demand.

As expected, we were able to show an expression of MCT1 and MCT4 in GFAP-positive cells as a marker of astroglial origin. Moreover, only MCT1 was expressed in many vWF-positive endothelial cells. In support of our results, an expression of MCT1 was already described in previous studies [4]. Lim et al. postulated a primary expression of MCT1 in endothelial cells [5].

Since MCT1 and MCT4 have an important contribution to the maintenance of glycolytic metabolism and consequently tumor cell survival, they depict an interesting aim for targeted therapy. The effectiveness of inhibiting the activity and expression of MCT1 was already shown using in vitro and in vivo GBM models [4]. Besides a reduction of tumor mass, a significant decrease in the number of blood vessels around the tumors was found in the treated group. In addition, a synergistic effect between the best described MCT inhibitor in literature, primary inhibiting MCT1, α-cyano-4-hydroxy-cinnamic acid (CHC) and temozolomide (TMZ) was observed, hence tumor cells could be sensitized to TMZ by this pretreatment. It has to be emphasized that the effect of the used inhibitor was dependent on the metabolic state of the cell line used, showing most efficiency particularly in more glycolytic cells [4]. In addition, MCT inhibition yielded promising results in the case of glioma stem cells (GCS) [6,40]. Since MCT1 was shown to be upregulated in GCS, the selective inhibition of MCT1 by AR-C117977 especially decreased the viability of GSCs, compared with that of non-GSCs. In fact, the treatment reduced cell viability at lower concentrations in GSCs than in non-GSCs [6]. A phase I clinical trial of the MCT1 inhibitor in patients with advanced cancer (NCT01791595) is already ongoing in the United Kingdom. Expected side effects, due to the ubiquitous expression of the MCTs, have not been detected to date [41,42].

Depending on the metabolic and molecular profile of the tumor (metabolic state, stemness and dormant properties), the administration of an antimetabolic drug targeting MCTs could be verified.

In this regard, our observation of a subgroup, showing a lower MCT1 expression at the edge of the tumor compared with the center combined with an induction of EMT, stemness and dormancy markers at the invasive front and a high regional concentration gradient of lactate becomes interesting. In this group, the response to a MCT1 inhibitor might be limited by the aggressive properties of the invasive front of the tumor, leading to tumor recurrence. On the other hand, tumors showing a more homogenous distribution of the markers might profit more from this targeted therapy. In the previous work of our group, we found the chemotherapeutic agent AT101 combined with TMZ to be an effective treatment strategy against dormant glioma cells in vitro [7]. Depending on the molecular profile, a combination of this therapy with metabolic targeting could be another treatment option. Our observation of an expression of MCT4, but not MCT1, in dormant cells could also imply that patients with a higher percentage of dormant tumor cells also profit from inhibiting this transporter.

Furthermore, our findings indicate a possible connection between the distinct regional gene expression patterns of MCT1, EMT, stemness and dormancy markers with the regional lactate concentration. Recent examinations using hyperpolarized carbon-13 MRSI could also show a significant correlation between the lactate to pyruvate ratio and MCT1 expression in breast cancer [43]. Definitely, an exclusive contemplation of the regional lactate concentration will not allow any conclusion to be drawn concerning the molecular features of the tumor. Nevertheless, the development of the MRSI technique delivers more and more insights into tumor metabolism. In future, it might be possible to use the information gained by MRSI for the evaluation of the use of an antimetabolic treatment.

In addition, MRSI can monitor the response to treatment, as was already demonstrated for the efficiency of TMZ in GBM in vitro and in vivo [44]. Concerning the MCT inhibitor CHC, MRSI analysis indicated distinct changes in the brain metabolite profiles, due to the application of the metabolic inhibitor, offering the option to monitor the therapeutic effects of this agent as well [42].

Of course, the number of patients included in the study was relatively small and surely limits the transferability of our results into a clinical context. The already known strong heterogeneity of the disease makes it even more difficult to detect patterns in the gene expression profile of the different markers.

Despite some limitations, this work provides a detailed look at the location of MCT1 and MCT4 in GBM, revealing clear connections with EMT, stemness and dormancy processes. In order to better understand the exact function and role of MCT1 and MCT4, further research is required. In particular, a further investigation regarding a connection with dormant cells as a further examination of the microenvironment would be of great interest.

## 4. Conclusions

Despite all research efforts, glioblastoma multiforme still remains an uncurbable disease. Especially, metabolic reprogramming to an aerobic glycolysis, as well as EMT, stemness and dormancy phenomenon play crucial roles in tumor progression and therapy resistance.

A consideration of the local distribution of associated markers revealed a significant downregulation of the lactate transporter MCT1 at the edge of the tumor, whereas the gene expression of the EMT marker β-catenin and the stemness markers KLF4 and OCT4 was induced at the edge of the tumor. A further analysis concerning the gradient of the marker expression from center to edge of the tumor revealed different subgroups with distinct expression patterns. A strong decrease in MCT1 expression and increase in EMT, stemness and dormancy marker expression at the edge of the tumor compared with the center correlated with a high lactate gradient between the different regions. The higher concentration of lactate in the center of the tumor compared with the edge was measured in vitro (lactate ELISA) and in vivo (MRSI). Beside revealing different cell types to be the source of MCT expression, a so far unknown relation between MCT4 and the dormancy phenomenon was found by immunofluorescence double-staining.

Since MCTs have become a promising target of therapy in GBM, our results point out that attention should be paid to the connection between metabolic reprogramming with EMT, stemness and dormancy phenomenon, which might contribute to the resistance against antimetabolic drugs.

The found subgroups with distinct patterns of gene expressions imply, that beside a molecular screening to evaluate a possible response to targeted therapy, a combination with agents against the described tumor promoting processes might be reasonable. In addition to the monitoring of therapy response, a further development of the MRSI technique might contribute to select patients with an assumable greater benefit from antimetabolic treatment in the future.

Altogether, understanding the mechanisms involved in generating the aggressive properties of glioblastoma multiforme is imperative to improve the disastrous prognosis of this disease. More research and innovation are urgently needed.

## 5. Materials and Methods

### 5.1. Tumor Specimens and Ethical Consideration

After obtaining the written informed consent of the donors, the GBM tissue samples were collected during surgery by the Department of Neurosurgery (University Medical Center Kiel, UKSH, Campus Kiel, Kiel, Germany). A sample was taken from the center and another from the edge of the vital portion of the tumor (Figure 7). The study was approved by the local Ethics Committee of the University of Kiel (D 444/18, date of approval 10 April 2018 and D 524/17, date of approval 30 August 2017) and is in accordance with the Helsinki Declaration of 1975 and its further amendments. All samples included were classified as WHO grade IV IDH wild-type by the Department of Neuropathology (University Medical Center Eppendorf, UKE, Hamburg, Germany). An overview of the clinical data is given in Table 1. If sufficient tissue was available, a variety of examinations were carried out on the tissue. In total, 8 patients received MRSI (one patient was included twice, with his primary and recurrent tumor, respectively), 10 patients were examined with lactate ELISA and 15 patients were examined with qRT-PCR. We were able to perform the immunofluorescence staining of 5 patients.

### 5.2. Multi Voxel Magnetic Resonance Spectroscopic Imaging

GBM patients underwent MRI and MRSI before surgery, using a 32-channel head coil at 3 Tesla (Achieva, Philips Medical Systems, Eindhoven, The Netherlands; point-resolved spectroscopy (PRESS) chemical shift imaging (CSI) with echo time (TE) = 288 ms; repetition time (TR) = 1500 ms; voxel size 10 × 10 × 10 mm, matrix depending on anatomy 12 × 14 – 32 × 32; averages 1).

The MRSI data was analyzed with a linear combination model (LCModel, Version 6.3-1J, Stephen Provencher Inc., Oakville, ON, Canada) to determine the lactate signal in different regions [45]. The areas were classified as “tumor”, meaning the contrast enhancing part of the GBM in T1-weighted images after intravenous contrast agent injection; as “edema”, meaning the surrounding T2-weighted hyperintense edema without contrast enhancement; as “normal appearing white matter” (NAWM), bordering the tumor edema; and as “contralateral”, meaning normal appearing white matter of the contralateral hemisphere. As expected, the spectral appearance varied between tumors and their location. Generally speaking, spectral quality was best on the contralateral side, and at a distance from the ventricles and skull. To calculate the mean lactate signal, up to three representative voxels were picked from each area, depending on the quality, size and location of the GBM and the edema.

### 5.3. Lactate ELISA

ELISA was performed following the manufacturer’s instructions with the L-lactate assay kit (ab65331, Abcam, The Netherlands). In detail, after the homogenization of individual center and edge GBM tissue pairs (10–20 mg tissue each) with the lactate assay buffer (ab65331, Abcam, The Netherlands), endogenous lactate dehydrogenase was removed using the deproteinizing sample preparation kit—TCA (ab204708, Abcam, Rozenburg, The Netherlands). Lactate concentrations (nmol/mg tissue) were assessed for each pair of center and edge samples in relation to internal lactate standard curves, respectively, in a microplate reader for optical density at 450 nm (GENios, TECAN, Zürich, Switzerland).

### 5.4. Quantitative Reverse Transcription and Real-Time PCR

RNA of individual GBM center and edge pairs, respectively, was isolated with the TRIzol^®^ reagent (Invitrogen, Carlsbad, CA, USA), digested by DNase (Promega, Madison, WI, USA); cDNA was synthesized using RevertAid™ H Minus Reverse Transcriptase (Thermo Scientific, Schwerte, Germany), and qRT-PCR was performed using TaqMan primer probes (Applied Biosystems, Foster City, CA, USA), as described before [7]: MCT1 (Hs-01560299_m1), MCT4 (Hs-00358829_m1), β-catenin (Hs-00172016_m1), glial acidic fibrillary protein (GFAP) (Hs-00157674_m1), glycerinaldehyde-3-phosphate-dehydrogenase (GAPDH) (Hs-99999905_m1), TATA-box-binding protein (TBP) (Hs-00920494_m1), vimentin (Hs-00185584_m1), krüppel-like factor 4 (KLF4) (Hs-00358836_m1), octamer binding transcription factor 4 (OCT4) (Hs-00999632_g1), sex determining region Y-box 2 (Sox2) (Hs-00602736_s1), ephrin receptor A5 (EPHA5) (Hs-00300724_m1), histone cluster 1 H2B family member k (H2BK) (Hs-00955067_g1), insulin-like growth factor-binding protein 5 (IGFBP5) (Hs-00181213_m1). Fluorescent data were converted into cycle threshold (*C*_t_) measurements. Gene expression was quantified relative to that of the house-keeping genes GAPDH and TBP, used as internal standard. Δ*C*_t_ values of each sample were calculated as *C*_t gene of interest_—([*C*_t GAPDH_ + *C*_t TBP_]/2). The inclusion of two house-keeping genes was chosen to make our data more reliable. Ratios of gene expressions between tumor edge and center were calculated via the ΔΔ*C*_t_ method as 2^−(Δ*C*^_t edge_^−Δ*C*^_t center_^)^ = n-fold. Concerning a calibrator sample, we just matched the experimental samples to determine the relative gene expression ratios separately, since matched pairs of samples were available. Different shades of grey were assigned to different ratios of gene expressions of the individual center and edge pairs, in order to make possible similarities according to the expression of the different markers more easily recognizable. A ratio of 1 indicating an equal expression in center and edge was coded as 30% grey. Lower n-fold expression values, i.e., expression at edge lower than in center were displayed with increasingly lighter shading with values close to 0 corresponding to white. Relative expression values > 1, i.e., expression at the edge was higher than in center, had increasingly darker shades of grey up to a 3- or higher fold expression, which was assigned the color black. The individual edge-center GBM pairs were then arranged in a ‘heatmap’.

### 5.5. Immunofluorescence

Cryostat sections of GBM tissues were prepared as described previously [7]. Cells were incubated overnight with the primary antibodies at 4 °C, followed by the secondary antibodies for 1 h at 37 °C. The nuclei were counterstained with 4’,6-diamidino-2-phenylindole (DAPI; Thermo Fisher Scientific, Waltham, MA, USA; 1:30,000, 30 min, room temperature), and the embedded slides were analyzed by fluorescence microscopy (AxioObserver.Z1; Carl Zeiss, Oberkochen, Germany). If the primary antibodies were derived from the same species, non-specific binding was blocked by F(ab) fragments derived from that species (1:1000, from Jackson ImmunoResearch, West Grove, PA, USA). Primary antibodies were omitted for negative controls. In combination with anti-MCT1 (1:500, mouse, sc-365501, Santa Cruz Biotechnology Inc., Dallas, TX, USA) and anti-MCT4 (1:150, mouse, sc-376101, Santa Cruz Biotechnology Inc.), the cells were stained with the specific EMT antibodies β-catenin (1:150, mouse, 651109, BD Transduction Laboratories, Franklin Lakes, NJ, USA) and vimentin (1:50, mouse, sc-6260, Santa Cruz Biotechnology Inc.), as well as the stem cell antibodies KLF4 (1:250, mouse, MA5-15672, Thermo Fisher Scientific), OCT4 (1:150, rabbit, #2750, Cell Signaling, Danvers, Massachusetts, USA), Sox2 (1:200, rabbit, sc-20088, Santa Cruz Biotechnology Inc.), the dormancy antibodies EPHA5 (1:400, rabbit, sc-927, Santa Cruz Biotechnology Inc.), H2BK (1:400, rabbit, orb184226, Biorbyt, Cambridge, UK) and IGFBP5 (1:400, rabbit, sc-13093, Santa Cruz Biotechnology Inc.), and the antibodies GFAP (1:500, mouse monoclonal, MAB3402, Millipore, Billerica, MA, USA) and von Willebrand factor (vWF) (1:1000, sc-53465, mouse monoclonal, Santa Cruz Biotechnology Inc.). The MCT markers were always stained first with Alexa Fluor 488-coupled secondary antibodies (green, 1:1000, donkey anti-mouse IgG, Thermo Fisher Scientific), the second secondary antibody detecting the cell-specific markers was donkey anti-rabbit IgG Alexa Fluor 555 (red, 1:1000, Thermo Fisher Scientific).

### 5.6. Statistical Analysis

The data were statistically analyzed using the GraphPad Prism 8.4^®^ software (GraphPad Software, San Diego, CA, USA). The multi-voxel MRSI results were analyzed using one-way-ANOVA with Tukey’s post hoc test. For further statistical analyses, two-tailed Student’s *t*-test for dependent samples was used. Correlations were calculated with the Pearson correlation index. Significance levels ranged between * *p* < 0.05, ** *p* < 0.01 and *** *p* < 0.001.

## Figures and Tables

**Figure 1 ijms-21-06254-f001:**
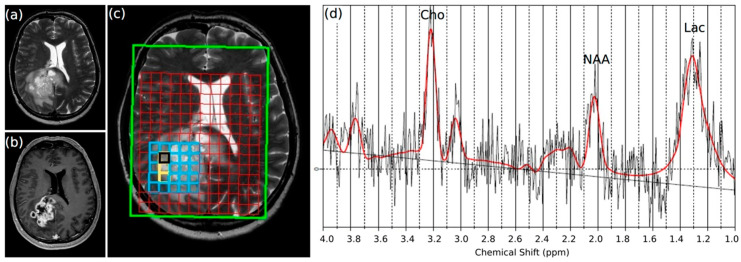
Representative magnetic resonance (MR) examination of a glioblastoma multiforme (GBM) patient. The same slice is shown in different weighted sequences. (**a**) T2-weighted image; (**b**) T1-weighted image after administration of the i.v. contrast agent Gadolinium; (**c**) T2-weighted image with the grid of the multi voxel magnetic resonance spectroscopic imaging (MRSI); the black square highlights the spectrum shown in part (**d**); (**d**) spectrum of a voxel in the contrast-enhancing part of the GBM analyzed with a linear combination model (LCModel). Besides an elevated choline-peak at 3.2 ppm (Cho) and a decreased NAA (*N*-acetyl aspartate)-peak at 2.0 ppm, an elevated lipid/lactate-peak (Lac) at 1.3 ppm was detected.

**Figure 2 ijms-21-06254-f002:**
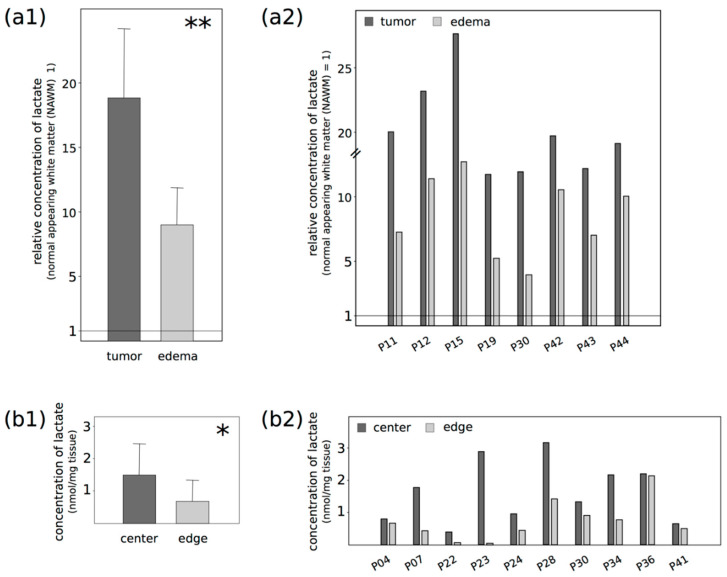
Comparison of lactate concentrations in different glioblastoma regions (P: patient/probe), as evaluated by (**a**) multi voxel magnetic resonance spectroscopic imaging (MRSI) and (**b**) lactate ELISA. (**a**) Based on the “normal appearing white matter (NAWM)” as reference (NAWM = 1), the relative lactate concentrations are given for the contrast-enhancing part of the tumor (“tumor”)—meaning in T1-weighted images after intravenous contrast agent injection—and for the edema without contrast enhancement (“edema”); (**b**) The lactate concentrations were determined in paired center and edge tumor samples and given as nmol/mg tissue. Both methods revealed a significantly different concentration of lactate comparing the examined regions. The relative lactate concentrations were significantly higher in the (**a**) contrast-enhancing part of the tumor than in the edema or the NAWM (** *p*: 0.006), and at the (**b**) tumor center compared with the edge (* *p*: 0.042). Statistically significant differences were determined by (**a**) one-way ANOVA or (**b**) two-tailed Student’s *t*-test. (**a1**) and (**b1**) show the average values of the determined concentrations, whereas (**a2**) and (**b2**) plot the values of each probe.

**Figure 3 ijms-21-06254-f003:**
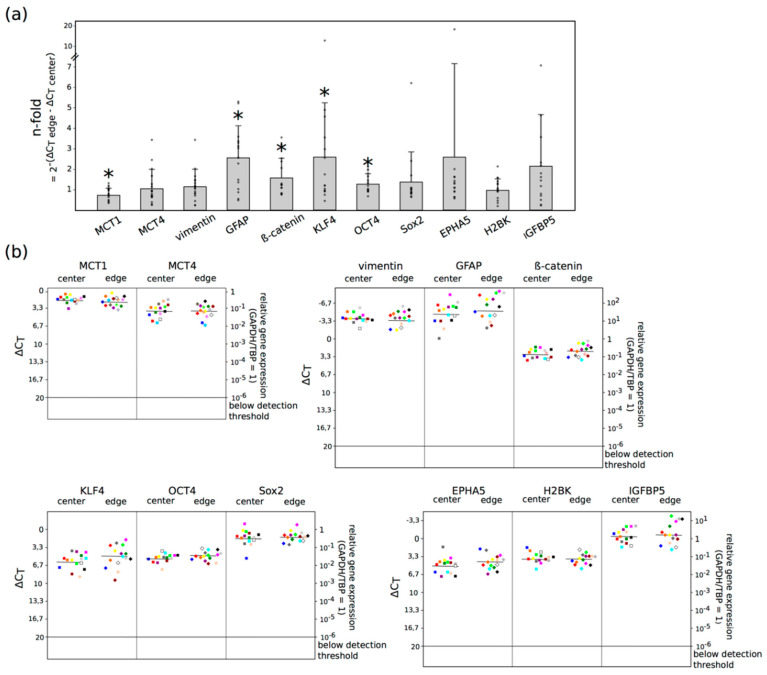
Expression of MCT (monocarboxylate transporter) 1, MCT4, EMT (epithelial to mesenchymal transition) markers (vimentin, β-catenin), stem-like cell markers [KLF4 (krüppel-like factor 4), OCT4 (octamer-binding transcription factor 4), Sox2 (sex-determining region Y-box 2)], dormancy markers [EPHA5 (ephrin receptor A5), H2BK (histone cluster 1 H2B family member k), IGFBP5 (insulin-like growth factor-binding protein 5)] and GFAP (glial acidic fibrillary protein) in pairs of center and edge glioblastoma samples was evaluated by quantitative reverse transcription PCR (logarithmic scale, a Δ*C*_t_ = 3.33 increase corresponds to a 10-fold decrease in gene expression). (**a**) Shows the n-fold changes of each marker between the center and the edge of the tumor (calculated with the average values of 2^−(^^Δ*C*^_t edge_
^−^^Δ*C*^_t center_^)^ = n-fold), small dots plot the n-fold value of each probe, whereas (**b**) plots the Δ*C*_t_ values itself. GAPDH (glyceraldehyde-3-phosphate dehydrogenase) and TBP (TATA-box-binding protein) served as internal standards. The expression of MCT1 was significantly lower (* *p*: 0.025), whereas GFAP (* *p*: 0.010), β-catenin (* *p*: 0.039), KLF4 (* *p*: 0.035) and OCT4 (* *p*: 0.026) showed a significant higher expression at the edge of the tumor compared with the center, as determined by two-tailed Student’s *t*-test. Furthermore, a slightly increased expression (more than 2-fold) of EPHA5 and IGFBP5 was detectable at the edge of the tumor compared with the center.

**Figure 4 ijms-21-06254-f004:**
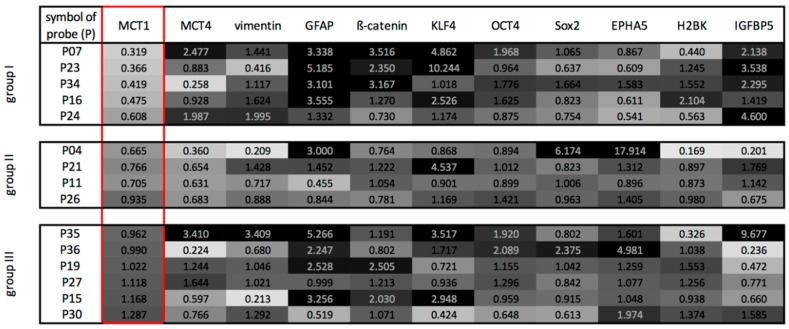
Heatmap-like analysis of gene expression ratios (calculated as 2^−(^^Δ*C*^_t cedge_
^−^^Δ*C*^_t center_^)^) between pairs of the edge and center glioblastoma samples equal expression at the edge and center pairs is assigned as 30% grey, lower expressions at the edge of the tumor is assigned an increasing lighter shading with a ratio close to 0 corresponding to white, and higher expressions at the edge of the tumor is assigned with increasing darker grey shades until 3 or higher-fold expression which is assigned as black. Mainly depending on the n-fold expression changes of MCT1, therefore outlined in red, three groups with distinct patterns of marker expression could be distinguished: group I showing a downregulation of MCT1 combined with an upregulation of GFAP and individual EMT-, stemness- and dormancy-associated markers (β-catenin, KLF4, IGFBP5) at the edge of the tumor—group II showed a slightly to moderate downregulation of MCT1 and MCT4 at the edge of the tumor, whereas the EMT-, stemness- and dormancy-associated markers were expressed heterogeneously and group III revealed a heterogenous pattern of all examined markers.

**Figure 5 ijms-21-06254-f005:**
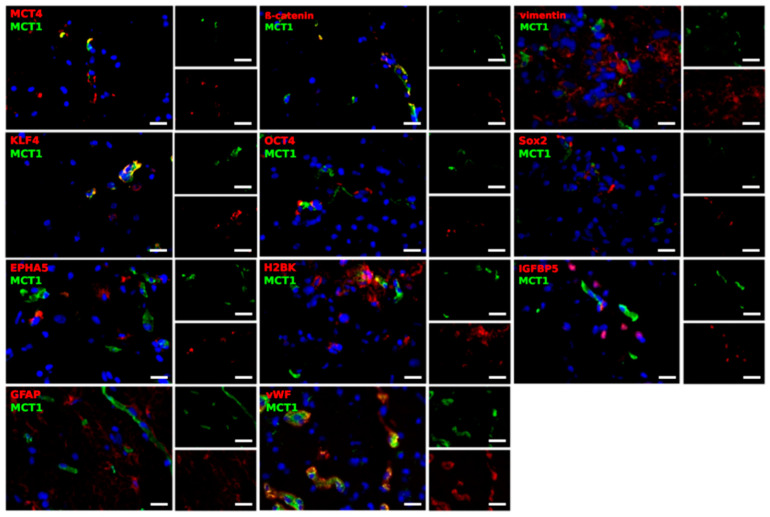
Representative co-staining of MCT1 with EMT markers (β-catenin, vimentin), stem-like cell markers (KLF4, OCT4, Sox2), dormancy markers (EPHA5, H2BK, IGFBP5) and GFAP, as well as vWF in pairs of center and edge glioblastoma samples as determined by immunofluorescence microscopy. MCT1 and MCT4 (green) are found in different combinations with cell-specific markers (red) in a complex pattern. Magnification 400×; bar = 20 μm.

**Figure 6 ijms-21-06254-f006:**
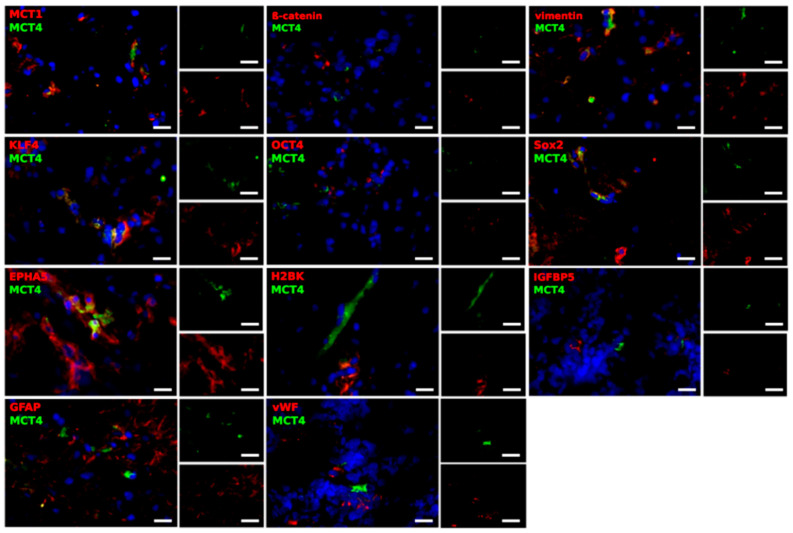
Representative co-staining of MCT4 with EMT markers (β-catenin, vimentin), stem-like cell markers (KLF4, OCT4, Sox2), dormancy markers (EPHA5, H2BK, IGFBP5) and GFAP, as well as vWF, in pairs of center and edge glioblastoma samples, as determined by immunofluorescence microscopy. MCT1 and MCT4 (green) are found in different combinations with cell-specific markers (red) in a complex pattern. Magnification 400×; bar = 20 μm.

**Figure 7 ijms-21-06254-f007:**
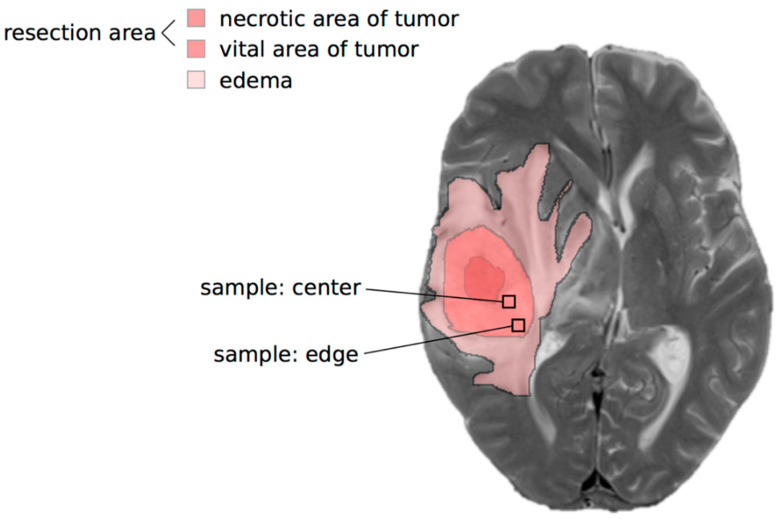
Typical topographic view of the glioblastoma resection area (vital area of the tumor (medium shade of red) and necrotic area (deepest shade of red)) and the surrounding edema (lightest shade of red), superimposed on an axial T2-weighted MR image (grey). Sampling areas from the center and edge of the vital parts of the tumor are indicated.

**Table 1 ijms-21-06254-t001:** Overview of the clinical data of the examined patients (P: probe). The age, gender, the time span of symptoms before diagnosis, the setting: primary or recurrent tumor, the time span, till a relapse was diagnosed, as well as isocitrate dehydrogenase (IDH), O(6)-methylguanine-DNA methyltransferase (MGMT) promoter and 1p19q status was regarded. Two patients were included twice with primary and recurrent tumor, respectively (P11_primary_ = P29_recurrence_; P06_primary_ = P35_recurrence_).

Symbol of Probe (P)	Age (Years)	Gender (Female ♀; Male ♂)	Symptoms Before Diagnosis (Days)	Primary Versus Recurrence	Time to Relapse (Months)	IDH Status	MGMT Status	1p19q Status
P01	50	♂	3	recurrence	6	wild-type	not methylated	co-deletion
P02	52	♂	28	recurrence	13	wild-type	not methylated	no co-deletion
P04	43	♀	11	primary		wild-type	methylated	no co-deletion
P05	66	♂	3	primary		wild-type	methylated	no co-deletion
P06	78	♂	1	primary		wild-type	methylated	no co-deletion
P07	62	♂	3	primary		wild-type	not methylated	no co-deletion
P11	63	♀	0	primary		wild-type	methylated	no co-deletion
P12	71	♂	3	primary		wild-type	methylated	no co-deletion
P15	59	♂	3	primary		wild-type	methylated	no co-deletion
P16	78	♂	3	primary		wild-type	not methylated	co-deletion
P19	73	♂	60	recurrence	3	wild-type	not methylated	no co-deletion
P21	56	♂	0	primary		wild-type	methylated	no co-deletion
P22	78	♂	2	primary		wild-type	not methylated	no co-deletion
P23	81	♀	180	primary		wild-type	methylated	no co-deletion
P24	57	♀	14	primary		wild-type	methylated	no co-deletion
P26	56	♂	180	primary		wild-type	not methylated	not measured
P27	52	♀	0	recurrence	17	wild-type	methylated	no co-deletion
P28	79	♂	0	primary		wild-type	methylated	not measured
P29	63	♀	0	recurrence	3	wild-type	methylated	no co-deletion
P30	63	♀	14	primary		wild-type	unclear	not measured
P34	63	♂	30	primary		wild-type	methylated	not measured
P35	79	♂	0	recurrence	15	wild-type	methylated	no co-deletion
P36	74	♂	21	primary		wild-type	not methylated	not measured
P41	69	♀	60	primary		wild-type	not methylated	not measured
P42	71	♂	30	primary		wild-type	methylated	not measured
P43	58	♀	2	primary		wild-type	not methylated	not measured
P44	75	♀	0	primary		wild-type	methylated	not measured

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
