# Peer review of "Intratumoral Distribution of Lactate and the Monocarboxylate Transporters 1 and 4 in Human Glioblastoma Multiforme and Their Relationships to Tumor Progression-Associated Markers"

_ijms, 2020, doi:10.3390/ijms21176254_

Round 1
Reviewer 1 Report
In this new version the quality of this manuscript has significantly improved. I would like to thank the authors that they have addressed all my former comments very detailed. I have no further comments and suggest that the manuscript is published based on the current version.
Reviewer 2 Report
The authors have fully and well addressed all concerns suggested. Despite the number of patients analyzed in this work is low, the findings obtained are important for increasing knowledge on the GBM biology. The results showed may also contribute to the development of targeted therapies. In my opinion, the revised manuscript is now suitable for publication in present form, including English language and style.
This manuscript is a resubmission of an earlier submission. The following is a list of the peer review reports and author responses from that submission.
Round 1
Reviewer 1 Report
The manuscript entitled:“Intratumoral distribution of lactate and the monocarboxylate transporters and in human glioblastoma multiforme and their relationships to tumor progression-associated markers” is overall of interest and the topic GBM is of significant importance considering the high malignancy of this neoplasm and the lack of effective curative treatments.
However, some major points should be addressed by the authors for the manuscript to be published, so that the data shown are consistent with the objectives and conclusions:
- The number of patients analyzed to carry out the molecular investigations reported in the manuscript should be specified in the text, as the robustness of the results obtained strongly depends on the patient number.
- In the title section 2.2 the authors report: “Two main GBM groups were distinguished based on their relative n-fold mRNA expression differences in the center and at the edge of the tumor”. How many patients were identified for each group?
- As regards the gene expression data, the authors should better show the qRT-PCR results in order to highlight more clearly the differences in gene expression obtained. The Figure 3 should be modified, showing gene expression data in fold change as histograms, for example.
- Considering that the gene expression data are relative, the authors should better specify which is the test sample and the calibrator sample in each qrt-PCR experiment for the calculation of the DDCt.
- Why did the authors choose the GAPDH gene as a house-keeping gene, considering that hypoxia is one of the main causes of GBM therapy resistance and that metabolic genes are highly variable and not very stable under hypoxic conditions? This represents a critical point for all gene expression results.
In addition, some minor points should be also addressed:
- The normalized averaged DCt values of all investigated markers reported in the text in lane 133 and later should be shown in a table. Why don't the authors report the fold changes?
- The asterisks shown in Figure 2 and the statistical significance of the data should be better explained.
- The authors show in the table 1 some patient clinical data about the IDH mutation status and the MGMT methylation status, but there is no mention of how these data may correlate with the results obtained, in particular with the distinct MCT expression profiles. So, the authors should specify whether there is a possible correlation between the patient's clinical data and the results obtained.
- The authors should better explain in the discussion section the possible clinical applications on GBM in particular for any targeted therapy, regarding the results obtained from this work.
- A conclusion section should be added in the manuscript.
Reviewer 2 Report
This excellent publication investigates the expression and distribution of MCT1 and MCT4 in glioblastoma, which is one of the "hot topics" in targeting GBM for imaging and treatment approaches. As was shown with other methods before, MCT1 and MCT4 are expressed in GBM. However, here MRI spectrometry is used to confirm these findings in vivo before surgical examination. The manuscript is well written and I have only some minor comments/suggestions to be addressed before final publishing:
In general: What is the value of this information for therapy decision and patient treatment. Please specifiy and discuss within the manuscript. If possible, please provide a patient treatment related rational in the abstract as well.
Results: Please provide MR images of the contrast-enhancing part of the GBM
Line 77: "lactate concentration of all patients" please provide all mentioned data (spectrum, lactate concentration etc.) for all patients
Line 89-91: please combine these sentences.
Figure 1: Consider to put abbreviations like Lac, Cho or NAA directly above the peak in the spectrum. Furthermore, why not showing multi voxel spectrum here?
Figure 2: “Sp” might not clear to the reader at this point. Please define in the caption. Please provide separated in supplemental material all patients data for all investigated regions as well!
Line 144-145: "Interesting results were..." please delete sentence.
Line 153-155: Rewrite to:Interestingly, comparing the lactate concentration in the center and edge of the tumor, the first group showed a slight or moderate difference while the second group exhibit the highest difference.
Line 156: "...statistically significant correlation..." Although not significant, please provide p values in supplement.
Figure 4: Consider to add text: "group 1" and "group 2"
Discussion:
Please discuss the value of your findings with regard to patient treatment or improvement of current therapies.
Line 248-251: Is there a therapy consequence out of this? Please discuss the "value for the patient"
Table 1:Please provide more details in the caption. E.g. what are the different grey tones belonging to? Reading the caption should provide enough information to understand it without reading the whole manuscript.
Why Sp 21=Sp11 etc. please define in table caption
Line 339: Please define "contrast enhanced part". MR contrast agent used or contrast in MR image?
Reviewer 3 Report
Kubelt et al. assessed lactate concentration and gene expression, including MCT1 and MCT4, at multiple sites in GBM. They also performed immunohistochemistry to assess co-expression pattern with other markers, such as EMT, and stemness. Unfortunately, these findings do not provide novel insights for GBM biology, unless the authors show functional role of MCT1 and MCT4 and functional interaction with other molecular markers.